# Aided Diagnosis Model Based on Deep Learning for Glioblastoma, Solitary Brain Metastases, and Primary Central Nervous System Lymphoma with Multi-Modal MRI

**DOI:** 10.3390/biology13020099

**Published:** 2024-02-05

**Authors:** Xiao Liu, Jie Liu

**Affiliations:** School of Computer and Information Technology, Beijing Jiaotong University, Beijing 100044, China; xiaoliu@bjtu.edu.cn

**Keywords:** brain tumor, classification, convolutional neural network, feature fusion

## Abstract

**Simple Summary:**

Diagnosing glioblastoma multiforme (GBM), solitary brain metastases (SBM), and primary central nervous system lymphoma (PCNSL) in malignant tumors of the central nervous system using multi-modal magnetic resonance imaging (MRI) is significantly important in helping physicians develop treatment plans and enhance patient prognosis. In this paper, MFFC-Net is developed and validated using deep learning methods to predict these three tumor categories from multi-modal MRI without the manual region of interest (ROI). MFFC-Net first uses a multi-encoder with DenseBlocks to extract deep features from multi-modal MRI. Then, the feature fusion layer fuses the deep information between different modalities and tissues. Finally, the spatial-channel attention module suppresses redundant new information and activates tumor classification-related features. Compared with radiomics models, MFFC-Net demonstrated higher accuracy. In addition, the results in the different sequences provide important references for future clinical work on MRI image acquisition. We believe that MFFC-Net has the potential to assist in the diagnosis and treatment of brain tumors in the future.

**Abstract:**

(1) Background: Diagnosis of glioblastoma (GBM), solitary brain metastases (SBM), and primary central nervous system lymphoma (PCNSL) plays a decisive role in the development of personalized treatment plans. Constructing a deep learning classification network to diagnose GBM, SBM, and PCNSL with multi-modal MRI is important and necessary. (2) Subjects: GBM, SBM, and PCNSL were confirmed by histopathology with the multi-modal MRI examination (study from 1225 subjects, average age 53 years, 671 males), 3.0 T T2 fluid-attenuated inversion recovery (T2-Flair), and Contrast-enhanced T1-weighted imaging (CE-T1WI). (3) Methods: This paper introduces MFFC-Net, a classification model based on the fusion of multi-modal MRIs, for the classification of GBM, SBM, and PCNSL. The network architecture consists of parallel encoders using DenseBlocks to extract features from different modalities of MRI images. Subsequently, an L1−norm feature fusion module is applied to enhance the interrelationships among tumor tissues. Then, a spatial-channel self-attention weighting operation is performed after the feature fusion. Finally, the classification results are obtained using the full convolutional layer (FC) and Soft-max. (4) Results: The ACC of MFFC-Net based on feature fusion was 0.920, better than the radiomics model (ACC of 0.829). There was no significant difference in the ACC compared to the expert radiologist (0.920 vs. 0.924, *p* = 0.774). (5) Conclusions: Our MFFC-Net model could distinguish GBM, SBM, and PCNSL preoperatively based on multi-modal MRI, with a higher performance than the radiomics model and was comparable to radiologists.

## 1. Introduction

Glioblastoma (GBM) is a malignant brain tumor formed as a result of mutations in the genetic material and epigenetic mechanisms driving continuous cell cycle progression and mitosis in brain cells, leading to abnormal energy metabolism that supports sustained growth [1]. Solitary brain metastases (SBM), on the other hand, refer to the highly malignant tumors that have spread to distant organs or tissues through the bloodstream [2]. Primary central nervous system lymphoma (PCNSL) is a rare and highly malignant non-Hodgkin lymphoma characterized by the malignant clonal proliferation of lymphocytes, including intracerebral lymphocytes and lymphocytes with central nervous system involvement [3]. Malignant tumors of the central nervous system (CNS) primarily consist of gliomas, meningiomas, pituitary adenomas, ventricular meningiomas, CNS lymphomas, and metastatic tumors. Among these, GBM is the most common malignant primary brain tumor, accounting for 77–81% of all primary malignant tumors of the CNS [4]. During the progression of their disease, between 20% and 40% of patients with systemic cancers will develop metastases [5]. PCNSL represents approximately 6% of intracranial malignancies [6]. CNS tumors are more prevalent in Northern Europe, and they also have a significant impact on countries like China, the United States, and India. The high incidence of CNS tumors in these regions represents a substantial health burden, emphasizing the need for effective strategies in terms of prevention, diagnosis, and treatment [7]. These three classes of brain tumors are malignant brain tumors that occur in the CNS, with typical clinical manifestations of elevated intracranial pressure and various neurological symptoms [8,9]. Due to the similarity of conventional MRI findings among the three (as shown in Figure 1), the three common malignant tumors occur in the central nervous system. These tumors share similar imaging characteristics, including central necrosis, irregular or garland-like enhancement of the tumor margins after contrast enhancement, and extensive edema surrounding the tumor [10,11]. Conventional MRI sequences and traditional medical image analysis methods can sometimes make it challenging to differentiate between these three tumor types, particularly for less experienced doctors. Studies have shown that when brain metastases appear as solitary lesions without a clear history of a primary tumor, the similarity of imaging features with high-grade gliomas can lead to misdiagnosis in approximately 40% or more cases [12]. Additionally, while GBM and PCNSL often exhibit distinct MRI manifestations, there are instances where differentiation becomes difficult. For example, atypical PCNSL tumors containing necrosis and hemorrhage may resemble GBM, while atypical GBM tumors without necrosis and with solid appearances may resemble PCNSL [13,14]. However, there are significantly different treatments for different tumors. Early selection of the most appropriate treatment option can greatly improve the prognosis. A portion of patients are forced to choose surgery/puncture due to lack of a clear diagnosis, causing unnecessary trauma and delaying the optimal treatment opportunity. Therefore, mastering accurate identification of the three types of tumors before treatment is of significant clinical importance for guiding clinical treatment, optimizing patient management, and improving patient prognosis.

With the advancement of medical imaging and computer information technology, Lambin et al. [15] proposed the concept of radiomics in 2012, and the analytical methods of radiomics have been rapidly developed and applied in medical imaging-related fields. Several researchers have used radiomics models to classify brain tumors [16,17]. They used manual or overview regions of interest (ROIs) for feature extraction and then constructed classification models based on machine learning (ML) modeling methods. However, describing ROIs is a very subjective, tedious, and time-consuming task. To address this issue, scholars have established numerous automatic segmentation models for brain tumor ROI annotation. These models include unsupervised modeling methods [18,19], supervised machine learning (ML) methods [20,21], and approaches that combine the strengths of both [22,23]. These methods, although capable of performing automatic brain tumor segmentation, require an image preprocessing process. Deep learning (DL) is a complex nonlinear regression method, which has developed into a new research direction in ML. By applying DL, data features can be extracted accurately, automatically, and efficiently, avoiding the errors that may be caused by manual segmentation, which can save manpower, financial resources, and time. It has been applied for the qualitative diagnosis, efficacy evaluation, and prognosis evaluation of many diseases of brain tumors [24,25,26,27]. A convolutional neural network (CNN) is the most commonly used classification model in deep learning. Due to their ability to directly learn the most relevant features related to brain tumor ROI, as well as their adaptive ability and nonlinear representation of data, CNNs have been widely used in multi-modal MRI-based brain tumor classification research.

In existing studies, researchers have focused on the construction of DL-based dichotomous models to complete the brain tumor-assisted diagnosis model [28,29,30], including the differentiation of GBM and SBM, as well as GBM and PCNSL. Few studies have been proposed on DL in the differential diagnosis of GBM, SBM, and PCNSL. The main reason for this may be that DL technology requires large-scale, high-quality, and standardized medical image data as training data input. However, it is tough to obtain a lot of imaging data on three types of brain tumors that meet the standards. Therefore, this study is based on the standardized medical imaging database of brain tumors established by Huashan Hospital, Fudan University, with the use of DL, to establish a triple classification prediction model for GBM, SBM, and PCNSL to achieve non-invasive and accurate diagnosis of three types of brain tumors before treatment, providing evidence for patient subsequent treatment and buying time.

This paper presents MFFC-Net, a multi-modal MRI fusion-based model for assisting in the diagnosis of three types of common and histologically similar malignant CNS tumors: GBM, SBM, and PCNSL. The key contributions of this work can be summarized as follows:DenseBlock-based parallel multiple encoders are proposed to extract features simultaneously from different sequences. This allows for comprehensive representation learning across various MRI sequences.A novel L1−norm feature fusion module is introduced to enhance the interrelated information between different tumor tissues. By improving the tumor characterization ability of the extracted features, the model achieves more accurate tumor classification.The model incorporates a spatial-channel self-attentive weighting operation on both the modal and fusion features. This operation dynamically adjusts the relationship between the weights of different channels, enhancing the model’s expressive ability and improving its overall performance.

By leveraging these contributions, MFFC-Net demonstrates promising potential for assisting in the diagnosis of GBM, SBM, and PCNSL, thereby aiding in the effective management and treatment of these malignant CNS tumors.

## 2. Patients

### 2.1. Patient Enrollment and MRI Scanning Parameters

Institutional review board approval (No. KY2021-066) was obtained from Huashan Hospital of Fudan University in this study. We accessed the relevant brain tumor medical imaging database for the period from February 2014 to November 2022 and obtained multi-modal MRI data from patients with pathologically confirmed GBM, SBM, and PCNSL. A total of 1225 patients with brain tumors were ultimately included in this study, including 419 patients with GBM, 412 patients with SBM, and 394 patients with PCNSL. By statistical analysis, we found no statistically significant differences in the gender and age of these patients, and the baseline characteristics of the enrolled patients are shown in Table 1.

Three MRI scanners were used, including Signa HDxT 3T (GE Healthcare, Milwaukee, WI, USA), Discovery MR750W 3T (GE Healthcare, Milwaukee, WI, USA), and Magnetom Verio 3T (Siemens Healthineers, Erlangen, Germany). Brain MRI scanning parameters are shown in Table 2.

### 2.2. Data Preprocessing

Before constructing the CNN model, all personal information included in the study was anonymized, and a series of standardized automatic pre-processing was performed on all MRI sequences. Firstly, we used SimpleITK (version 2.1.1.1) to resample CE-T1WI to the same resolution as T2-Flair, ensuring that the spacing, origin, and direction of CE-T1WI were consistent with T2-Flair. Then, the mean μ and standard deviation σ of the voxel intensities were statistically calculated and, according to μ and σ, we adjusted the voxel intensities to [μ−3σ,μ+3σ]. After that, we utilized Advanced Normalization Tools (ANTs) (https://github.com/ANTsX, accessed on 5 October 2022) to register the T2-Flair to CE-T1WI of the same case. Finally, maximum external square clipping of the brain mask for each MRI slice was performed, the matrix was rescaled to 240×240, and each slice was normalized to [0, 255] before being fed into the DL model.

## 3. Method

### 3.1. Classification Network Construction

We built a multi-modal MRI classification network with feature fusion named MFFC-Net, as shown in Figure 2, and the Structure of MFFC-Net is shown in Figure A2. Data input of the MFFC-Net was defined as multi-modality MRI images input $\left(x_{t1ce},x_{flair}\right)$. Firstly, $\;x_{t1ce}$ gets the low-level features through the convolution layer of 3×3. Then, $f_{0}^{t1ce}$ proceeded high-level features extraction and acquired $f_{1}^{t1ce}$, $f_{2}^{t1ce}$ and $f_{3}^{t1ce}$ using a single modality encoder. Similarly, $x_{flair}$ obtained $f_{0}^{flair}$, $f_{1}^{flair}$, $f_{2}^{flair}$, $f_{3}^{flair}$ by parallel identical encoding methods. After that, the L1−norm strategy was used to act $onf_{n}^{t1ce}$ and $f_{n}^{flair}\left(n=1,\;2,\;3\right)$ achieving the fusion feature $f_{fusion}$, and *n* was the number of DenseBlocks of the single modality encoder. High-level features $f_{3}^{flair}$, $f_{3}^{t1ce}$ and fusion feature $f_{fusion}$ were spliced together through concat operation as the final feature of the encoder $f_{all}=concat\left(f_{3}^{t1ce},f_{3}^{flair},f_{fusion}\right)$. The MFFC-Net decoder, on the other hand, consisted of two fully connected layers and a classification Soft-max layer. Feature $f_{all}$ was utilized as the input and gained a one-dimensional feature after passing through the full convolutional (FC) layer, and then the final classification result was obtained by Soft-max.

The MFFC-Net encoder consisted of two independent modality encoders connected in parallel to perform feature extraction on CE-T1WI and T2-Flair sequence images, respectively. Each encoder was composed of three DenseBlocks connected in series, as shown in Figure 2. Each DenseBlock contained 5 convolutional layers. In addition to feature extraction, DenseBlock (as shown in Figure 3a) was able to integrate the previous inputs. Finally, an average pooling layer was used to reduce the number of dimensions and parameters of the model. Eventually, the two independent modality encoders obtained the features.

The MFFC-Net decoder, on the other hand, consisted of FC and Soft-max layers. After the feature $f_{all}$ was fed into the FC layer, the final classification result was output through the final Soft-max layer. The predicted outcome for each patient was represented by the average of the predicted values of all slices containing the tumor. The loss function L was defined as Equation (Equation 1),
(1)$$L=-\frac{1}{N}\sum_{i=1}^{N}\sum_{l=1}^{3}y_{i}^{l}log\left(P\left(f_{all};\theta \right)\right)$$
where *i* was the index of MRI slices, *N* was the number of MRI slices, *l* was the index of pathologically confirmed tumor type (label), *L* was the number of label indexes (L=3 in this paper, corresponding to GBM, SBM, and PCNSL, respectively), $y_{i}^{l}$ was the label corresponding to the slice, θ was the parameter set of the CNN model, and *P* was the output probability.

Since the parallel independent modality encoders only extracted texture and detail information of the respective modalities of CE-T1WI and T2-Flair, they did not extract complementary information between the two modalities (e.g., information on the relative position and grayscale contrast between different tissues of the tumor). Inspired by Li et al. [31], we introduced the L1−norm fusion layer (as shown in Figure 3b) into this study to fuse the features of two modalities at different scales. The high-level features were unified into 30×30×128 by a resize operation. The fusion feature map $f_{fusion}^{m}$ was calculated using Equation (Equation 2),
(2)$$f_{fusion}^{m}\left(x,y\right)=\sum_{i=1}^{K}\omega_{i}\times \Phi_{i}^{m}\left(x,y\right)$$
where $\omega_{i}$ was the feature weight, ${\left\{\Phi_{i}^{m}\right\}}_{i=1}^{K}$ was the feature map, and m=128, K=2 was the number of modality indices for multi-modal images.
(3)$$\omega_{i}\left(x,y\right)=\frac{\hat{C_{i}}\left(x,y\right)}{\sum_{n=1}^{K}\hat{C_{n}}\left(x,y\right)}$$
where $\hat{C_{i}}$ was the activity level map. Alternatively, the L1 parametrization of $\Phi_{i}^{1:M}$ can be used as the activity level metric for the feature map and was obtained from Equation (Equation 4),
(4)$$C_{i}\left(x,y\right)={∥\Phi_{i}^{1:M}\left(x,y\right)∥}_{1}$$
where the activity level map $\hat{C_{i}}$ was obtained by Equation (Equation 5),
(5)$$\hat{C_{i}}\left(x,y\right)=\frac{\sum_{a=-r}^{r}\sum_{b=-r}^{r}C_{i}\left(x+a,y+b\right)}{{\left(2r+1\right)}^{2}}$$
where the average operator size was r×r and the fusion strategy of this paper was set as r=2.

### 3.2. Model Training

The hardware environment for the model development was a workstation with Ubuntu 20.04, Intel Xeon CPU, NVIDIA GTX 3090 GPU, and 64GB RAM. MFFC-Net constructed through pytorch11.10 and python3.9. Adam optimizer trained the network for 100 periods with an initial learning rate $\eta =10^{-4}$ and a weight decay $\omega =10^{-5}$. MFFC-Net was constructed based on T2-Flair and CE-T1WI with 5-fold cross-validation. We partitioned the dataset into five-fold cross-validation before conducting model training and testing. In each iteration, we used four-fifths of the data for training and reserved one-fifth for testing. We repeated this process until all the data had been utilized for testing purposes. To evaluate the importance of each MRI sequence, we also trained the other two networks in the case of a single sequence to evaluate the results obtained by the different modalities in the classification task.

### 3.3. Evaluation Indicators

We constructed a ML-based radiomics model by the radiomics approach we proposed in [32]. Subsequently, the MRI images of all cases in the dataset were independently assessed by three physicians specializing in diagnostic neuro-oncology radiology. The first physician had 5 years of experience, the second had 10 years, and the third had 20 years of experience. They individually examined the images and classified the tumors as GBM, SBM, or PCNSL based on their expertise. The three radiologists were referred to as junior, senior, and expert radiologists, respectively.

We compared the effectiveness between the radiomics model, the DL model, and the clinician model by a confusion matrix. We utilized the ROC curve to determine the optimal threshold point, also known as the “Cutoff”. This Cutoff value is subsequently employed to classify the classification results, thus facilitating the classifier in achieving its optimal performance. Results were assessed by the following metrics: accuracy (ACC), positive predictive value (PPV), sensitivity (SEN), specificity (SPE), area under the curve (AUC), F1-score, and net reclassification improvement (NRI). The evaluation indicator formulas are in Appendix A.

### 3.4. Statistical Analysis

This paper utilizes IBM SPSS Statistics 26.0 for statistical analysis and implements a significance level of *p* < 0.05 to identify statistical differences. Various tests are employed in the study, including the Mann–Whitney U test, Fisher exact test, Pearson chi-square test, and DeLong test. The Mann–Whitney U test is utilized to evaluate differences in age among two sampling groups, while the Fisher exact test is used to analyze gender differences among patients. The Pearson chi-square test is employed to construct a model and examine diagnostic differences between radiologists. Lastly, the DeLong test is implemented to compare the performance of ROC curves and assess the significance of the AUC values. Overall, the paper ensures the integrity of the analysis by utilizing appropriate statistical tests and applying a strict significance level.

## 4. Results

Table 3 and Figure 4 show the results of brain tumor trichotomies for the radiomics and DL models. Among all radiomics models constructed based on a single sequence, the CE-T1WI-based model (CR-Model) obtained the highest ACC of 0.810. Compared to the CR-Model, the multi-modality radiomics model (MR-Model) increased the ACC by 0.19. In the DeLong test, no significant difference in AUC was identified between the CR-Model and MR-Model (0.859 vs. 0.873, *p* = 0.208). The DL model also showed results consistent with the performance of the radiomics model in the results obtained for single sequences. The CE-T1WI-based model (CC-Net) obtained the highest ACC in the single sequence DL models (ACC of 0.841 and AUC of 0.877). In contrast, the ACC of the MR-Net with multi-modal was significantly higher than that of the CC-Net (0.890 vs. 0.841, *p* = 0.021). Although MFFC-Net was not significantly different in ACC compared to MC-Net, the MFFC-Net AUC was 0.26 higher and significantly different than MC-Net (0.942 vs. 0.916, *p* = 0.032) and F1-score was 0.029 higher (0.919 vs. 0.890).

We compared the classification results of GBM, SBM, and PCSNL obtained by MFFC-Net with DenseNet [33], SENet [34], and EfficientNetV2-S [35]. The results are presented in Table 4. We observed that while both DenseNet and SENet achieved an AUC exceeding 0.90, it was still slightly lower compared to EfficientNetV2-S, which got an AUC of 0.938. However, our method demonstrated some improvement in ACC, PPV, SEN, SPE, and F1-score compared to EfficientNetV2-S, despite not showing a significant difference (*p* = 0.512 with the DeLong test) in terms of AUC.

Compared to clinician diagnoses (as shown in Figure 5), the MFFC-Net achieved excellent performance with no significant difference from expert radiologists in the ACC (0.920 vs. 0.924, *p* = 0.774).

As demonstrated in Figure 5, the MFFC-Net exhibited outstanding performance when compared to clinician diagnoses. Specifically, our model achieved an accuracy (ACC) score of 0.920, which was comparable to that of expert radiologists with no significant difference between the two (0.920 vs. 0.924, *p* = 0.774). These results further validate the effectiveness and reliability of our proposed MFFC-Net as a diagnostic tool for identifying brain tumors.

To visualize the classification weights of the DL model, we plotted the gradient-weighted class activation mapping (Grad-CAM) to visualize the DL-based model of the DL model for a more intuitive understanding of the ROIs of the DL model, as shown in Figure 6. The red areas correspond to high scores in the tumor category. We found that the MFFC-Net model focuses more on the tumor region.

## 5. Discussion

In this paper, we developed and tested the MFFC-Net for GBM, SBM, and PCNSL classification. MFFC-Net extracted high-level features for T2-Flair and CE-T1WI parallel encoding, respectively. Then, a feature fusion layer was constructed to enhance the interrelationship information between different tumor tissues and suppress redundant features. After completing the tumor classification task by convolution and Soft-max layers, the deep-fused features were concatenated together. Furthermore, we compared the diagnostic efficacy of radiomics models, DL models, and radiologists.

Among the single-sequence classification models (including the FR-Model, CR-Model, FC-Model, and CC-Model), the efficacy of the CE-T1WI sequence-based models was superior to that of the T2-Flair sequence-based models (SEN of radiomics models: 0.810 vs. 0.728, SPE of radiomics models: 0.905 vs. 0.865, AUC of radiomics models: 0.859 vs. 0.797; SEN of DL models: 0.840 vs. 0.750, SPE of DL models: 0.920 vs.0.875, AUC of DL models: 0.877 vs. 0.818) consistent with our clinical work experience [36,37]. It is proved that CE-T1WI can more visually reflect the cellular anisotropy, neovascularization, degree of blood–brain barrier disruption, and infiltration of surrounding tissues in brain tumors [38,39]. However, the weak correlation between edema region features and tumor type (as shown in Figure 6) may have led to the poor performance of the T2-Flair-based classification model in this task.

As for the multi-modal MRI-based classification models (including MC-Net, and MFFC-Net), the MC-Net model based on multi-modal MRI had better diagnostic efficacy than either model based on single MRI (SEN of radiomics models: 0.829 vs. 0.728, SPE of radiomics models: 0.915 vs. 0.865, AUC of radiomics models: 0.873 vs. 0.797; SEN of DL models: 0.889 vs. 0.750, SPE of DL models: 0.945 vs. 0.875, AUC of DL models: 0.916 vs. 0.818). The result is consistent with the performance of radiomics-based classification of brain tumors reported by Bae et al. [40]. To some extent, these also reflected the potential significance and value of multi-modal MRI in radiomics and DL model construction and clinical application. Furthermore, by Figure 6 we found that both CC-Net and MC-Net enable the network to focus on the tumor area. MFFC-Net, on the other hand, significantly reduces the weight of non-tumor regions (the weight map of normal brain regions is more blue-oriented). AUC of MFFC-Net based on fusion feature was significantly better than MC-Model (0.942 vs. 0.916, *p* = 0.038), which fully indicates that the fusion of deep features, which can better characterize the tissue relationship of tumors, suppresses redundant features, reduces the variance of prediction, and decreases the generalization error [41]. In addition, the proposed feature fusion layer can improve the classification ability of the DL model.

In addition, the ACC of MFFC-Net was significantly better than the junior radiologist (0.940 vs. 0.782, *p* < 0.001) and senior radiologist (0.940 vs. 0.879, *p* = 0.017). There was no statistically significant difference between MFFC-Net and expert radiologists in ACC (0.940 vs. 0.943, *p* = 0.775). In a sense, the ACC of diagnostic imaging depends heavily on the clinical work experience of radiologists and needs to be improved by long-term clinical practice. And our MFFC-Net can effectively compensate for the lack of diagnostic experience of junior radiologists. It is not the only one: Shin et al. [42] developed a classification model using ResNet50 for multi-modal MRI and achieved AUCs of 0.889 and 0.835 in the internal and external test sets, respectively. These results were generally consistent with those obtained by radiologists, who achieved AUCs of 0.889 and 0.857, respectively. It can be seen that our MFFC-Net can help them improve the differential diagnosis of three types of brain tumors and gain time for patients’ subsequent treatment. The model also prevents patients from being forced to opt for surgery or puncture due to the inability to confirm the diagnosis, which causes unnecessary harm, by assisting diagnosis in a non-invasive manner.

## 6. Limitations

Firstly, MFFC-Net only used two sequences (T2-Flair and CE-T1WI) to construct the model. More sequences need to be incorporated in further studies to improve the model classification performance. Secondly, the model needs to be validated more extensively on a larger independent dataset, and multi-center studies are still needed to validate the robustness and generalizability of MFFC-Net. Thirdly, this study utilizes cross-validation as a means to assess the model’s performance. However, it lacks testing on an independent test group, which poses a limitation. Hence, we recommend incorporating an independent test group in future studies to validate the model’s generalizability and stability. In addition, When comparing diagnostic accuracy between MFFC-Net and clinical doctors, it is important to note that the number of participating physicians was limited to only three. This may have introduced a level of subjectivity to the results, which represents a limitation of this study. Finally, a limitation of this study is that it only considers GBM, SBM, and PCNSL, while excluding the inclusion of other histologically different types of brain tumors. We plan to address this in our future research.

## 7. Conclusions

In this paper, we propose a novel deep learning model, MFFC-Net, for the three classifications of GBM, SBM, and PCNSL. The model is based on parallel multi-channel encoding and feature fusion, which allow us to analyze images of different modalities effectively and efficiently. Our experiments demonstrate that MFFC-Net can achieve non-invasive and accurate diagnosis of GBM, SBM, and PCNSL before treatment, providing an invaluable tool for medical professionals. Furthermore, we conducted a comparison between our model’s results and those of radiologists’ diagnoses. The findings indicate that MFFC-Net can assist less experienced medical professionals and improve diagnostic accuracy, which is crucial for ensuring timely and effective treatment decisions. We believe that MFFC-Net has the potential to advance the diagnosis and treatment of brain tumors and can be extended to other medical applications as well.

## Figures and Tables

**Figure 1 biology-13-00099-f001:**
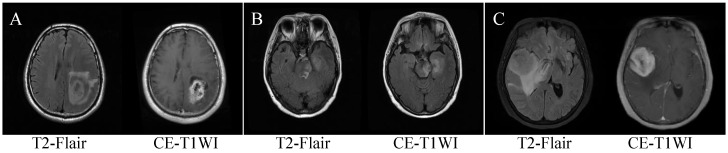
MRI images of three types of brain tumors. (**A**) T2-Flair and CE-T1WI of GBM; (**B**) T2-Flair and CE-T1WI of SBM; (**C**) T2-Flair and CE-T1WI of PCNSL.

**Figure 2 biology-13-00099-f002:**
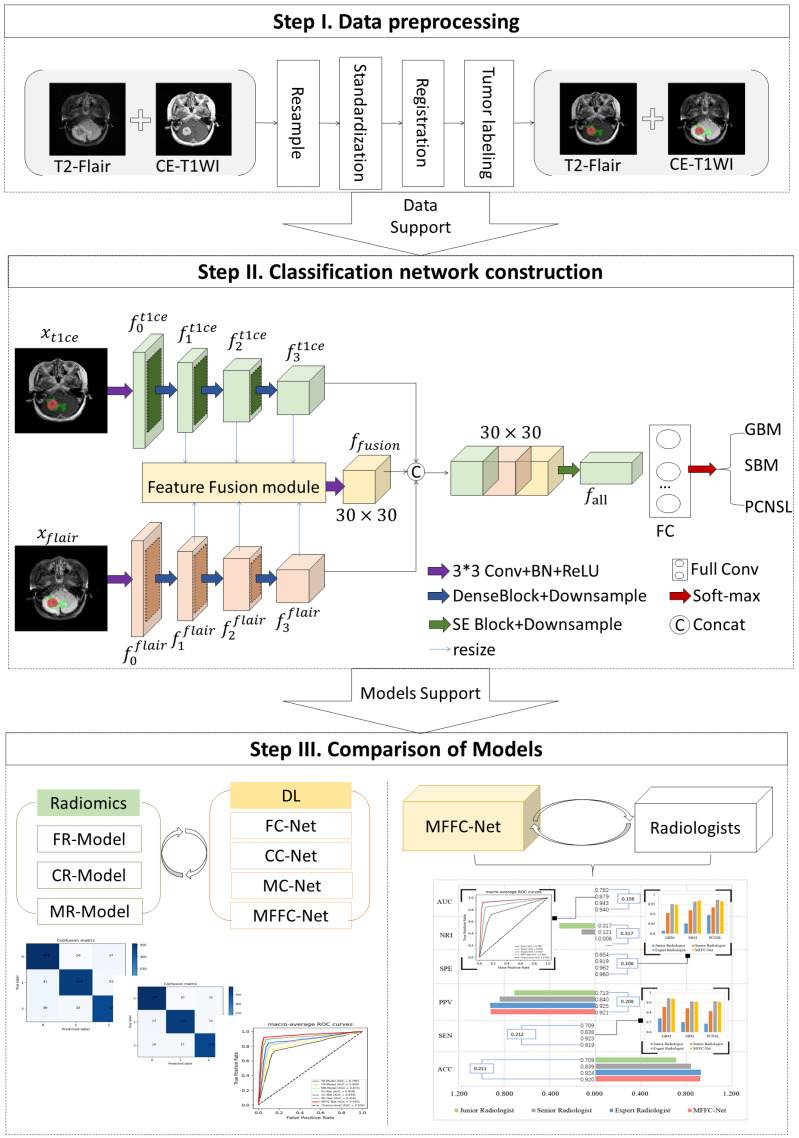
The schematic workflow of MFFC-Net. The first step includes data processing including resampling, registration, and normalization. The second step uses CE-T1WI and T2-FLAIR to construct a DL classification model based on feature fusion. The third step evaluates the classification results of the three brain tumors’ radiomic models, CNN models, and radiologists.

**Figure 3 biology-13-00099-f003:**
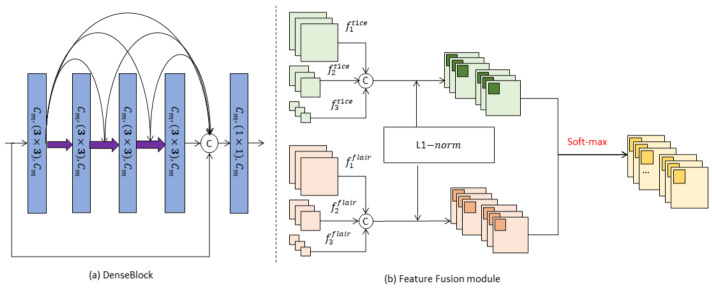
Network architecture of the modules, (**a**) DenseBlock, (**b**) Feature Fusion module.

**Figure 4 biology-13-00099-f004:**
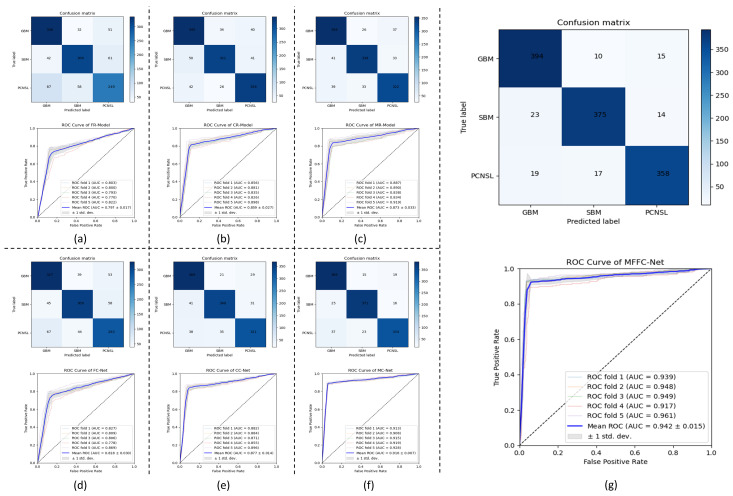
The Confusionmatrix and ROCs of the models. (**a**–**g**) are the confusion matrices and ROC for different models.

**Figure 5 biology-13-00099-f005:**
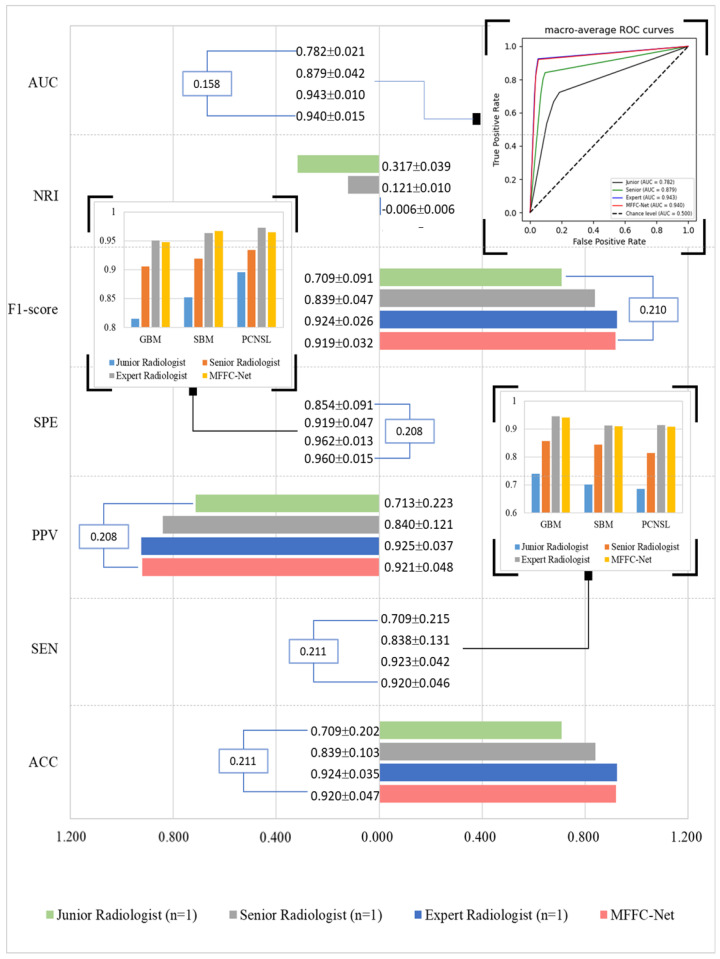
Comparisonresults of MFFC-Net and diagnostics of radiologists. The graph shows from bottom to top the ACC, SEN, PPV, SPE, F1-Score, NRI, and AUC of diagnostic results obtained by the junior radiologist, senior radiologist, expert radiologist, and MFFC-Net. *n* is the number of radiologists. The blue box shows the maximum boost results between MFFC-Net and radiologist diagnosed (see Appendix C original data).

**Figure 6 biology-13-00099-f006:**
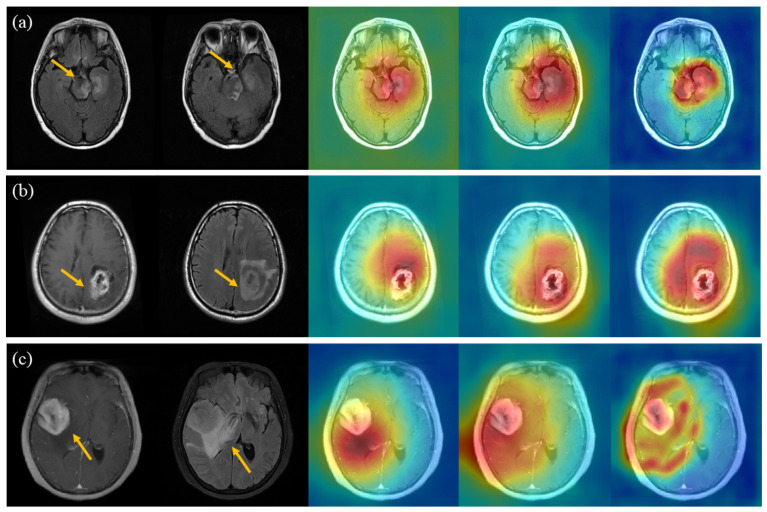
Multi-modalMRI (first and second columns) and the corresponding Grad-CAM obtained by the model (CC-Net, MC-Net, and MFFC-Net, respectively), the yellow arrows refer to brain tumors. (**a**) GBM, (**b**) SBM, (**c**) PCNSL.

**Table 1 biology-13-00099-t001:** Baseline characteristics of enrolled patients.

Characteristic	All	GBM	SBM	PCNSL	*p*-Value
Age (year)	53 ± 13	53 ± 11	56 ± 12	55 ± 13	0.319
Gender					
Male	671	228	208	235	0.215
Female	554	191	204	159	
Total	1225	419	412	394	

Footnotes: age is shown as the mean value ± standard deviation; all others are shown as the number of patients. The raincloud plot of age is shown in Figure A1; p≤0.05 represents the significance of comparisons. Abbreviations: GBM = glioblastoma; SBM = solitary brain metastases; PCNSL = primary central nervous system lymphoma.

**Table 2 biology-13-00099-t002:** Multi-modal MRI sequence parameters.

MRI Scanner	T2-Flair	CE-T1WI
Signa 3T	TR = 6880 ms; TI = 1850 ms;	TR = 1650 ms; TI = 720 ms;
	TE = 140 ms; Matrix = 288 × 192;	TE = 23.7 ms; Matrix = 288 × 192;
	FOV = 240 × 240 mm^2^;	FOV = 240 × 240 mm^2^;
	Thickness = 5 mm;	Thickness = 5 mm;
	Interval = 1.5 mm	Interval = 1.5 mm
Discovery MR750W 3T	TR = 8000 ms; TI = 2100 ms;	TR = 2992 ms; TI = 869 ms;
	Matrix = 256 × 256;	Matrix = 320 × 320;
	FOV = 240 × 240 mm^2^;	FOV = 240 × 240 mm^2^;
	Thickness = 5 mm;	Thickness = 5 mm;
	Interval = 1.5 mm	Interval = 1.5 mm
Verio 3T	TR = 9000 ms; TI = 2500 ms;	TR = 2000 ms; TI = 857 ms;
	TE = 102 ms; Matrix = 256 × 190;	TE = 17 ms; Matrix = 256 × 168;
	FOV = 201 × 230 mm^2^;	FOV = 201 × 230 mm^2^;
	Thickness = 5 mm;	Thickness = 5 mm;
	Interval = 1.5 mm	Interval = 1.5 mm

Abbreviations: MRI = magnetic resonance imaging; T2-Flair = T2 fluid-attenuated inversion recovery; CE-T1WI = contrast-enhanced T1-weighted imaging; TR = repetition time; TI = inversion time; TE = echo time; FOV = field of view.

**Table 3 biology-13-00099-t003:** Results of brain tumor classification for the radiomics and DL models.

Methods	ACC	PPV	SEN	SPE	F1-Score	AUC
FR-Model	0.730 ± 0.172	0.729 ± 0.201	0.728 ± 0.210	0.865 ± 0.087	0.727 ± 0.145	0.797 ± 0.017
CR-Model	0.810 ± 0.121	0.811 ± 0.137	0.810 ± 0.141	0.905 ± 0.051	0.809 ± 0.0.84	0.859 ± 0.027
MR-Model	0.829 ± 0.105	0.830 ± 0.124	0.829 ± 0.131	0.915 ± 0.048	0.829 ± 0.076	0.873 ± 0.033
FC-Net	0.750 ± 0.155	0.750 ± 0.167	0.750 ± 0.164	0.875 ± 0.082	0.749 ± 0.107	0.818 ± 0.030
CC-Net	0.841 ± 0.086	0.842 ± 0.107	0.840 ± 0.112	0.920 ± 0.032	0.841 ± 0.070	0.877 ± 0.014
MC-Net	0.890 ± 0.052	0.891 ± 0.083	0.889 ± 0.085	0.945 ± 0.023	0.890 ± 0.061	0.916 ± 0.077
MFFC-Net	0.920 ± 0.047	0.921 ± 0.048	0.920 ± 0.046	0.960 ± 0.015	0.919 ± 0.032	0.942 ± 0.015

T2-Flair = T2 fluid-attenuated inversion recovery; CE-T1WI = contrast-enhanced-T1 weighted imaging; ACC = accuracy; SEN = sensitivity; SPE = specificity; AUC = area under the ROC curve; F1-score = F1-score; FR-Model = T2-Flair radiomics-based model; CR-Model = CE-T1WI radiomics-based model; MR-Model = Multi-modal radiomics-based model; FC-Net = T2-Flair-based network; CR-Model = CE-T1WI-based network; MR-Net = Multi-modal-based network; MFFC-Net = Multi-modal-based feature fusion network.

**Table 4 biology-13-00099-t004:** The results of MFFC-Net and other state-of-the-art methods.

Methods	ACC	PPV	SEN	SPE	F1-Score	AUC
DenseNet	0.886 ± 0.046	0.887 ± 0.045	0.885 ± 0.042	0.943 ± 0.012	0.885 ± 0.029	0.913 ± 0.010
SENet	0.906 ± 0.048	0.907 ± 0.058	0.905 ± 0.063	0.953 ± 0.021	0.906 ± 0.050	0.930 ± 0.013
EfficientNetV2-S	0.918 ± 0.054	0.919 ± 0.060	0.918 ± 0.057	0.959 ± 0.032	0.918 ± 0.401	0.938 ± 0.015
MFFC-Net	0.920 ± 0.047	0.921 ± 0.048	0.920 ± 0.046	0.960 ± 0.015	0.919 ± 0.032	0.942 ± 0.015

## Data Availability

Unavailability of data due to confidentiality agreement restrictions.

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
