# Peer review of "Aided Diagnosis Model Based on Deep Learning for Glioblastoma, Solitary Brain Metastases, and Primary Central Nervous System Lymphoma with Multi-Modal MRI"

_biology, 2024, doi:10.3390/biology13020099_

Round 1
Reviewer 1 Report
Comments and Suggestions for Authors
1. Write the novelty of the methodology clearly.
2. Compare the proposed the methodology with latest papers.
3. Write in detail the statistical analysis done by the authors.
4. Explain the different layers of classification network.
5. How much is the trainable parameters and non-trainable parameters of the deep learning network?
6. Conclusion is very short. Rewrite it.
7. Why the authors have chosen MFFC-NET? What are its advantages over other existing methods?
Author Response
We thank the reviewer for the valuable comments and suggestions, which we have revised and improved according to your requirements, and we will respond to them line by line in the following.

Reviewer 2 Report
Comments and Suggestions for Authors
The manuscript by Liu and Liu describes the DL-based tool MFFC-Net to discriminate between three specific types of brain tumours, with discrimination quality comparable with expert radiologists.
The authors concentrated only on three distinct types of brain tumours including glioblastomas, lymphomas and solitary metastases, not testing the MFFC-Net on other histologically distinct subtypes.
The limited mathematical description of the MFFC-Net tool has been provided, without the code of the developed DL-based model and the results of classification provided.
The Supplementary Materials are not uploaded in pair with the manuscript.
To ensure the reproducibility of the developed solution, the authors must improve the manuscript extensively.
Major issues.
[1]
According to the “National Brain Tumour Registry of China (NBTRC) statistical report of primary brain tumours diagnosed in China in years 2019–2020” (Xiao et al. 2023 Lancet Reg Health West Pac), primary brain tumours represent more than 100 histologically distinct subtypes, with the list not limited by glioblastomas, lymphomas and solitary metastases.
However, the authors did not provide any clear storyline in the “Introduction” section that specified the importance of the discrimination between the trio of tumour types selected.
Please rewrite the “Introduction”, providing the readers with the reasonable details motivating the creation of the DL-based tool to identify the only three types of brain tumours, excluding other histological variants.
[1a]
In the “Introduction” section, please identify incidence rates for histologically distinct subtypes of brain tumours, including glioblastomas, lymphomas, solitary metastases and other types.
[1b]
In the “Introduction” section, whether Han populations have similar or distinct incidence rates for glioblastomas, lymphomas, solitary metastases and other types of brain tumours, as compared with other world populations (US, Canada, EU and others).
[1c]
In the “Introduction” section, please identify all ethiopathological differences between described three and other types of brain tumours (in terms of primary/secondary origin; cells of origin; tumour site; MRI signal specificity; etc).
The comparison of brain tumour types motivating the selection of glioblastomas, lymphomas and solitary metastases must be supported by the table included into the primary text.
[1d]
In the “Introduction” section, please identify the reasons of not including into this study histologically distinct types of brain tumours other than glioblastomas, lymphomas and solitary metastases.
In the “Discussion” section, please identify the non-inclusion of histologically distinct types of brain tumours other than glioblastomas, lymphomas and solitary metastases as a critical limitation for this study.
For references, please see the following sources
Tabassum et al.. Radiomics and Machine Learning in Brain Tumors and Their Habitat: A Systematic Review. Cancers (Basel). 2023;15(15):3845. doi:10.3390/cancers15153845
Xiao et al. National Brain Tumour Registry of China (NBTRC) statistical report of primary brain tumours diagnosed in China in years 2019-2020. Lancet Reg Health West Pac. 2023;34:100715. doi:10.1016/j.lanwpc.2023.100715
[2]
Demographic and clinical characteristics must be visualized as Raincloud plots with jittering, the Median, IQR boxes and CI95 intervals visualized for each quantitative characteristic like age, between control and affected samples.
[3]
The Figure 2 does not provide the clear linear representation of the developed pipeline and was not refereed properly in the “2.2. Data preprocessing”, “3.1. Classification network construction” and “3.3. Evaluation indicators” subsections of the manuscript.
The representation of the whole workflow must be improved greatly to ensure clear representation of the analysis workflow, identifying clear correspondence between the text details and the Figure 2 elements.
[3a]
In particular, please provide clear correspondences between the diagram elements depicted in separate boxes as ANN parts at the step II “Classification network construction”.
[3b]
It is also possible to use the Y-axis as a global time axis discriminating the timeline of pipeline steps and substeps performed.
[3c]
Please improve the size of the texts and the quality of the scheme on the Fig. 2.
[4]
The created DL-based models were not tested using brain tumours other than glioblastomas, lymphomas and solitary metastases.
In the “Discussion” section, please clearly identify this critical limitation of this study.
[5]
In the “Discussion” section, please provide the reasons of not reporting F1 scores for DL-based and human-based classification models tested, as it was suggested by Zhang et al.(2023).
Zhang et al. An integrative non-invasive malignant brain tumors classification and Ki-67 labeling index prediction pipeline with radiomics approach. Eur J Radiol. 2023;158:110639. doi:10.1016/j.ejrad.2022.110639
[6]
The Table 3 “Multi-modal MRI Sequence Parameters” has been provided without any description of the data format reported.
[7]
The authors did not describe the method applied to deal with uncertainties of the final classification result obtained by soft-max.
[8]
While the strategy of results validation using the human-based scoring is possible, the authors must provide the reasons of using the specific pattern of validation in the “Limitation” subsection of the manuscript.
[8a]
Was a train-test subsampling tactics used in the provided work?
If not, then why?
Please clarify the design of the study clearly.
[9]
The Results section must describe all results reported on the Figures 3 and 4 in a clear sequential way.
[10]
The data reporting must be improved extensively.
[10a]
The data for ACC, SEN, PPV, SPE, NRI, and AUC depicted on the Fig. 3 were presented without the uncertainty intervals assessed by cross-validation.
On the Fig. 3, please depict the ACC, SEN, PPV, SPE, NRI, and AUC results as means and SDs or medians and IQRs.
[11]
The reported AUC curves evidence that curves based on Radiologist reports were constructed using limited numbers of data points.
[11a]
To ensure the high quality of the report, please report the data underlying the AUC ROC curves in a form of the supplementary xls/csv files.
[11b]
Please also increase the size of the Figure panels depicting AUC and ROC curves on the Figure 4 and on the Fig. 3.
[11c]
For each tumour assessed, please report individual scores provided by each radiologist and each model estimated in a form of the supplementary xls/csv file.
[11d]
On the Fig. 4, please depict the uncertainty boundaries for all ROC curves reported.
[12]
The confusion matrices provided on the Figure 4 must be reported in pair with the ROC points on the corresponding ROC curves specific for those confusion matrices.
[13]
No comparisons of the developed tool with other solutions were provided in the Discussion section.
Please improve or clarify in the “Limitation” subsection.
[13a]
In particular, no comparisons with SOTA solutions were provided.
Please improve or clarify in the “Limitation” subsection.
[14]
No Supplementary Material Part 1 and 2 were provided in pair with the manuscript.
Please improve.
[14a]
The code of the solution must be reported.
Otherwise, please provide the reasons of not publishing the code in the “Limitation” subsection.
Comments on the Quality of English LanguageThe storyline in the Introduction must be clarified and linearized.
Author Response
Thank you very much for your valuable comments and suggestions on our article, we have thoroughly revised the manuscript, and we will respond to your review comments one by one below.

Reviewer 3 Report
Comments and Suggestions for Authors
Dear Authors,
The manuscript seems to be balanced, well-structured and written in readable English.
This is an interesting manuscript dealing with the current topic of decision making in medicine based on conventional MRI combined with machine learning. The topic of the manuscript is original and of interest to the scientific community, and the techniques used in the work are state of the art.
The introduction covers appropriately the topic of the paper and the reason for the research performed is clearly justified.
The methodology is clearly presented and the data, based on concrete tables are suggestive. The data presented and discussed support the conclusions offered by the paper.
However, I have some suggestions which can help you to improve the work, please find them below:
1. In the discussion, the authors focus on two metrics: AUC and ACC. However, at least two other metrics calculated by the authors are very important in the context of the research: SEN (sensitivity) and SPE (specificity). The authors should expand the discussion to include an analysis of these metrics.
2. The other aspect that should be included in the discussion is the analysis of the results presented in the literature in relation to similar research.
3. The authors should present formulae for the error metrics presented in Table 3.
4. The title of Table 3 is incorrect.
5. The sentence on lines 65-66 should be rewritten.
Author Response

(The authors gave the same response as above.)

Round 2
Reviewer 2 Report
Comments and Suggestions for Authors
While most problems of the manuscript were revised by the authors effectively, a number of issues must be revised too.
[1]
In the text of the manuscript, the authors have not provided the future readers with a clear answer for the results validation strategy used in this study.
Nevertheless, the authors claimed in the answers for the initial review point 10a that “As a result of implementing 5-fold cross-validation, all data points were employed as test data.”
To ensure the clear comprehension of the validation strategy,
[1a]
In the “3. Method” section, the authors must explain whether the training and test sets were formed before the cross-validation step or not formed, with the cross-validation step to be a single suboptimal procedure employed by the authors.
[1b]
If no training and test sets were predefined before the cross-validation step, then the authors must indicate this limitation clearly in the “Limitation” subsection of the manuscript, indicating that only the cross-validation strategy was used in this study, without testing on the special testing group.
[2]
Despite the claim that “there was no necessity to compute uncertainty intervals subsequent to grouping results of cross-validation”, the data for ACC, SEN, PPV, SPE, NRI, and AUC for the MFFC-Net depicted on the Fig.5 must be provided with the uncertainty intervals assessed by the cross-validation.
On the Fig. 5, please depict the ACC, SEN, PPV, SPE, NRI, and AUC results as means and SDs or medians and IQRs.
[3]
The authors must include into the main text of the manuscript their description of the softmax criterion provided in the answers for the initial review (point 7).
[4]
The authors must report F1 scores (harmonic means of Prec and Rec) in the main text of the manusript, since these are not a mere weighted average of precision rate and recall rate or a mere reconciled mean of precision rate and recall rate.
[5]
The numbers of radiologists per each group of junior, senior, and expert radiologists are still unclear.
[5a]
Please revise the sentences in the subsection “3.3. Evaluation indicators”, indicating the numbers clearly for each group.
[5b]
Please indicate the numbers of radiologists scoring the data as the critical limitation of the study in the relevant “Limitation” subsection.
[5c]
In the caption for the Fig. 5, please indicate the numbers of radiologists per group directly (in example, as n=1)
[6]
The Figure 3 caption does not correspond to the Figure panel.
Please revise the Caption.
Comments on the Quality of English Language
Language issues
[P2-P1-L12] “CNS tumors … are affected in China…” – Please improve
The caption for the Fig. 5 is still unclear. Please revise, using plain English.
Again, please revise fuzzy wordings across the text.
Author Response

(The authors gave the same response as above.)

Round 3
Reviewer 2 Report
Comments and Suggestions for Authors
The critical issues have been resolved by the authors successfully.
Minor corrections are required only.
[1]
In the “3.3. Evaluation indicators” [LL219-222], please add the mention of the F1 score.
Comments on the Quality of English Language
Language must be checked thoroughly.
[2]
In the “Simple Summary”, the initial sentence is fuzzy and must be revised extensively:
“GBM, SBM, and PCNSL in central nervous system malignant tumours through multi-modal MRI has(?)...” – Please clarify the clause
– Please do not use undefined acronyms
– must be “have”
– “GBM, SBM, and PCNSL … have implications for assisting (?) physicians” – please clarify the meaning of “assisting”
[3]
In the “Simple Summary”,
“…MFFC-Net achieved better results” – please name a single most important indicator.
For example, “…MFFC-Net demonstrated higher accuracy”
[4]
In the “Abstract” [P1-P2-LL1-2],
“Diagnosis … played …” – Better to use “plays”
And so on...
Author Response

(The authors gave the same response as above.)
